# 4D LATENT WORLD MODEL FOR ROBOT PLANNING

## ABSTRACT

Learned world models are emerging as a powerful paradigm in robotics, offering a promising path toward task generalization, long-horizon planning, and flexible decision-making. However, prevailing approaches often operate on 2D video sequences, inherently lacking the 3D geometric understanding necessary for precise spatial reasoning and physical consistency. Recent work has begun to inject 3D signals into video world models (e.g., depth and normals), improving spatial understanding but still operating on surface-level projections that can struggle under occlusion and viewpoint changes. To overcome this limitation, we introduce the *4D Latent World Model*, which learns to predict the evolution of a scene's 3D structure within a structured sparse voxel latent space, conditioned on observations and textual instructions. Our representation encodes the scene holistically and can be decoded into diverse 3D formats (e.g., 3D Gaussian Splatting), enabling a more complete and physically consistent scene understanding. This 4D latent world model serves as a planner, generating future scenes that are translated into executable actions by a goal-conditioned inverse dynamics model. Experiments demonstrate that our model generates futures with superior visual quality, physical consistency, and multi-view coherence compared to state-of-the-art video-based planners. Consequently, our full planning pipeline achieves superior performance on complex manipulation tasks, exhibits robust generalization to novel visual conditions, and proves effective on real-world robotic platforms. Our website is available at https://iclr2026-4553.github.io.

## 1 INTRODUCTION

Learning a general-purpose agent that can solve a wide range of real-world tasks has always been a central goal in robotics. However, progress is constrained by the scarcity of large-scale, task-diverse, and interactive robotic data required to train such agents. As a result, recent work has focused on policy-based agents that directly map observations to actions (Lillicrap et al., 2015; Chi et al., 2023; Zhao et al., 2023; Xiong et al., 2021). Although these end-to-end policies can perform well in narrow, well-instrumented settings, they commonly fail to generalize under even modest distribution shifts, such as changes in lighting, viewpoint, or the composition of unseen tasks.

An alternative paradigm is to learn a dynamics model that predicts the consequences of actions, enabling planning and improving generalization. Traditionally, dynamic models have been used extensively in various robotic tasks such as locomotion and manipulation, enabling effective planners like model predictive control (MPC) to operate on top of them (Garcia et al., 1989; Mayne et al., 2000; Qi et al., 2025). Recent work in robot learning revives this approach by learning video generation-based world models from large scale datasets and combining them with planners or inverse dynamics modules (Du et al., 2023b; Yang et al., 2023). Such models predict how the environment will evolve conditioned on text or task specifications. This makes decision-making more interpretable and flexible, facilitates long-horizon planning, and improves generalization to unseen tasks and environments. However, video-based world models are inherently 2D and operate in pixel space, resulting in physical inconsistencies with the real 3D world and limiting accurate spatial understanding. This limitation becomes especially problematic in fine-grained manipulation tasks where accurate 3D cues are essential (Zhu et al., 2024; Ke et al., 2024).

The necessity for true 3D reasoning becomes apparent in high-precision manipulation. Imagine a robot attempting a task that requires precise physical interaction, such as inserting a key into a lock or a pen into a narrow holder. Even slight misalignment in depth or angle can cause failure. Such

Figure 1: Our 4D latent world model integrates multi-view images and text instructions to forecast future 3D dynamics, enabling robots to plan and execute tasks that require precise 3D understanding.

tasks demand more than recognition of visual appearance: they require an accurate understanding of 3D geometry, object pose, and spatial relationships. This fundamental challenge exposes a key limitation of modern robot learning.

Modeling the dynamics of a scene directly in 3D, however, is a challenging task. Traditional 3D representations, such as point clouds and meshes, preserve geometry but lose rich visual detail necessary for semantic understanding. Photorealistic representations like Neural Radiance Fields (NeRFs) (Mildenhall et al., 2021) or 3D Gaussian Splatting (Kerbl et al., 2023) better capture appearance, but are computationally intensive and not easily amenable to dynamic modeling. A common compromise is to predict RGB video along with depth and normals Zhen et al. (2025), which provide partial 3D cues but still reduce to surface-level projections, leaving them vulnerable to occlusions and viewpoint shifts. This leads to a question, *Can we build a model that inherently simulates dynamic 3D structures of the world?*

In this paper, we propose a **4D latent world model** for robot planning. Following the success of Latent Diffusion Models (Rombach et al., 2022) which utilize spatially-aware 2D feature maps rather than unstructured 1D global latents, we adopt a structured 3D latent representation (Xiang et al., 2025) for 3D scenes. Specifically, we encode the scene into a sequence of sparse voxel grids where active voxel holds a compact feature vector. The *grid latent* design allows us to maintain explicit 3D spatial biases, while benefiting from the computational efficiency and semantic abstraction of a low-dimensional latent space. Based on the structured 3D latents, our model learns the dynamics for the 3D scene and generates plausible future latents conditioned on current observations and text instructions. Unlike methods restricted to surface maps or videos, our latent captures holistic 3D information of the scene that can be decoded into various formats, such as point clouds or 3D Gaussian Splatting. This approach enables our world model to achieve a more complete understanding of 3D structures and generate futures with superior physical and spatial consistency. This detailed 3D information is then leveraged by a goal-conditioned inverse dynamics model, which translates the generated futures into precise robot actions and is especially effective for fine-grained, 3D-aware tasks. In summary, our contributions are as follows:

i) We introduce a 4D latent world model that predicts future 3D structures conditioned on current observations and text goal, achieving high visual quality, physical consistency, and robust viewpoint generalization.

ii) We propose a planning framework that leverages our model's detailed 3D predictions as geometrically rich goals for an inverse dynamics controller, enabling precise and spatially-aware manipulation.

iii) Experiments demonstrate that our method outperforms state-of-the-art robot world models in both generation quality and downstream robotic task performance, including strong zero-shot generalization to various visual changes, and effective transfer to a real-world robot task.

## 2 RELATED WORK

**General Purpose Embodied Models.** A dominant paradigm in robotic learning and embodied agents has been the development of large multitask policies that directly map sensory inputs to output actions. Through the collection of large-scale multi-task embodied and robotics datasets, such models (Reed et al., 2022; Lee et al., 2022; Huang et al., 2023; Zitkovich et al., 2023; Kim et al., 2024; Barreiros et al., 2025; Hou et al., 2025; NVIDIA et al., 2025; Black et al., 2024) are able to

solve tasks across many environments. However, there are two large challenges with constructing such general-purpose policies across many environments. First, the action space across environments is often misaligned, with existing works requiring careful action tokenization (Reed et al., 2022), and second, small changes in the environment cause policies to fail. To circumvent these issues, our work focuses on learning a 3D model of the world and then planning on top of the model to act in the environment. This approach allows us to use a shared underlying 3D state of the world to transfer across environments. At the same time, by learning the more complex task of modeling the 3D dynamics of the world, we are able to effectively generalize across many environmental changes.

**Generative World Models for Planning.** Learned world models have recently been explored for robot planning, often through video prediction from a single viewpoint (Janner et al., 2022; Ajay et al., 2022; Li, 2023; Ajay et al., 2023; Du et al., 2023a; Ko et al., 2023; Yang et al., 2023; Li et al., 2023; He et al., 2023; Alonso et al., 2024; Chen et al., 2024; Ubukata et al., 2024; Bar et al., 2025; Qi et al., 2025; Xie et al., 2025a). For example, UniPi (Du et al., 2023b) frames planning as generating a video trajectory, which improves interpretability but lacks explicit 3D structure, leading to inconsistencies under occlusion or viewpoint change. To address this, hybrid approaches such as TesserAct (Zhen et al., 2025) extend video models to predict future depth and normal maps, providing stronger spatial priors for manipulation. However, these methods are fundamentally 2.5D and operate in pixel space, which remain surface-level projections that struggle to maintain full multi-view coherence. In contrast, our method models dynamics directly in a 3D latent space, which enables inherent 3D modeling rather than relying solely on 2.5D projections, ensuring consistent multi-view rollouts and providing geometrically grounded subgoals.

**3D Dynamics and Planning with Explicit Geometry.** A parallel line of research learns dynamics over structured 3D representations such as point clouds, meshes, or NeRF-like fields, enabling physical simulation or relational reasoning for manipulation tasks. These include behavior-primitive dynamics for stowing (Chen et al., 2023), point-cloud relational planning (Huang et al., 2025), and deformable-object digital twins (Jiang et al., 2025). Similarly, graph-based dynamics have been applied to elasto-plastic manipulation (Shi et al., 2023; 2022) and latent relational planners (Huang et al., 2024), while others utilize compositional NeRFs for multi-object scenes (Driess et al., 2023). While these methods succeed in specific domains, they typically rely on object-centric factorizations, pre-defined primitives, or task-specialized graph structures. In contrast, our approach learns a holistic latent 3D scene representation. It aggregates multi-view geometry into a unified state, supports 4D rollouts directly in latent space, and decodes into diverse 3D formats (point clouds or multi-view images rendered from 3D Gaussians). This formulation allows our model to jointly model dynamics and planning in a unified framework while maintaining flexibility across tasks, without requiring predefined object primitives or action parameterizations.

## 3 FORMULATION OF LATENT WORLD MODELING

### 3.1 PROBLEM FORMULATION

Our goal is to build a 4D world model that learns the dynamics of a 3D environment over time. We formalize it as a conditional generator $g(\boldsymbol{o}_{t+1}, ..., \boldsymbol{o}_{t+T}|\boldsymbol{o}_t, a)$. Here, given the state of the 3D scene $\boldsymbol{o}_t$ at time $t$ and an action $a$, the model predicts a sequence of future 3D scene states $\{\boldsymbol{o}_{t+1}, ..., \boldsymbol{o}_{t+T}\}$ over a horizon $T$.

In practice, the complete 3D scene $\boldsymbol{o}_t$ is not directly observable. Instead, it is only seen through partial observations $\{o_t^{(i)}\}$, such as RGB or depth images from multiple cameras in real-world setups, or renderings from simulated viewpoints. These observations must be geometrically consistent, as they all describe the same underlying 3D structure $\boldsymbol{o}_t$. The action $a$ can range from a low-level control signal to a high-level semantic instruction. In this work, we focus on text-based instructions that specify the desired evolution of the agent and the environment. In our implementation, language instructions serve as the high-level action input that guides the latent rollout, while the inverse dynamics module produces the low-level robot commands as absolute joint positions.

Prevailing world modeling methods are primarily based on video generation, predicting sequences of 2D frames (sometimes augmented with depth and normals) from a single viewpoint $g^{(i)}(o_{t+1}^{(i)}, ..., o_{t+T}^{(i)}|o_t^{(i)}, a)$. A straightforward extension to 3D is to train separate world models for each viewpoint and then fuse their outputs. However, such designs do not naturally support true

4D world modeling. Instead, we introduce a *4D latent world model* that directly addresses the key requirements:

- **3D consistency**: By encoding the complete 3D scene at timestep $t$ into a single holistic latent representation $z_t$, our model ensures that predictions across views adhere to the same underlying 3D structure.
- **Multi-view reasoning**: The shared latent aggregates information from multiple observations, allowing cues from one view to inform predictions in others.
- **Flexible generalization**: The latent can be decoded into diverse explicit 3D formats (e.g., point clouds, 3D Gaussians), allowing the framework to adapt to novel viewpoints and various scene representations.

Together, these properties enable a unified 4D latent world model that predicts future latent states, $g(z_{t+1}, ..., z_{t+T} | z_t, a)$. The latent $z_t$ is designed to be decodable into various explicit 3D representations, such as point clouds or 3D Gaussians, which allows use to obtain any desired observation $o_t^{(t)}$ decoded from the state.

### 3.2 3D LATENT FOR SCENE REPRESENTATIONS

Our world model requires a 3D latent representation $z$ that is both compact enough for efficient dynamic modeling and expressive enough to capture the fine details of the complete 3D structure. Traditional representations, such as meshes, point clouds, or SDFs, often lack photorealism, while modern representations, like NeRFs (Mildenhall et al., 2021) or 3D Gaussians Kerbl et al. (2023), are computationally expensive to generate directly at every timestep.

To balance efficiency and expressivity, we adopt a structured, sparse latent representation inspired by SLAT (Xiang et al., 2025). Our latent scene representation $z_t$, is defined as a set of sparse voxel features: $z_t = \{(\boldsymbol{p}_i, \boldsymbol{f}_i)\}_{i=1}^{L}$. Here, within a discretized $N \times N \times N$ grid of the 3D scene, $\boldsymbol{p}_i \in \{0, 1, ... N-1\}^3$ denotes the 3D coordinate of one of the $L$ active voxels, and and $\boldsymbol{f}_i \in \mathbb{R}^d$ is a feature vector encoding local geometry and color. This representation balances structural information with latent compression. Compared to a standard dense 3D grid at resolution $N = 64$ requiring $64^3$ elements, our structured voxel latent mostly utilize a sparse set of approximately $L \approx 8000$ active voxels, each carrying a compact feature $d = 8$ in our settings. This is similar to the design of 2D Latent Diffusion Models, where the latent space preserves spatial topology ($H \times W$) but compresses the channel dimension for efficient generative modeling. This latent representation is connected to 2D multi-view observations with an encoder-decoder framework.

**Encoding from images to 3D latent.** To construct the latent $z_t$ from multi-view images, a pretrained DINOv2 encoder extracts patch-level embeddings. Then these 2D embeddings are unprojected in the 3D voxel grid. For each voexel, the unprojected DINOv2 features from multi-view images will be averaged to an embedding, then a sparse encoder $\mathcal{E}$ to produce the latent features $\boldsymbol{f}_i$.

**Decoding from 3D latent to images.** To get back to a renderable scene, a sparse decoder $\mathcal{D}$ maps each latent voxel feature $\boldsymbol{f}_i$ to a set of $K$ 3D Gaussians $\{(\boldsymbol{o}_i^k, \boldsymbol{c}_i^k, \boldsymbol{s}_i^k, \alpha_i^k, \boldsymbol{r}_i^k)\}_{k=1}^{K}$, which can be rendered into images from arbitrary viewpoints or be converted into a point cloud. This establishes a mapping from the 3D latent $z_t$ to observation $o_t^{(i)}$.

We use the pre-trained 3D encoder–decoder from TRELLIS, which was trained using RGB reconstruction losses (L1, D-SSIM, LPIPS) to supervise the 3D Gaussians. This encoder-decoder architecture bridges raw visual perception (2D images) and a structured internal 3D world state ($z_t$). With this representation in place, the next step is to learn temporal dynamics in latent space.

## 4 4D LATENT WORLD MODEL

We propose a *4D latent world model* to predict the dynamics of 3D scenes. The model generates future 3D structures conditioned on current observations and textual instructions. With the ability to model 3D dynamics, it can serve as a planner for robot manipulation tasks, and when combined with an inverse dynamics module, it converts predicted 3D futures into executable robot actions. An overview of the framework is shown in Fig. 2.

Figure 2: **4D latent world model for robot planning.** The model reconstructs a 3D latent from multi-view images. A 4D latent world model then predicts future latents conditioned on the current state and a text instruction, using a Single Dynamics Model for coarse structural changes and a Latent Generator for detailed features. The predicted latents are decoded into explicit 3D formats such as point clouds or rendered views, which are subsequently used by a goal-conditioned inverse dynamics model to produce robot actions.

## 4.1 CONDITIONED 3D LATENT SEQUENCE GENERATION

We formulate 4D world modeling as a conditional generator in latent space: $g(z_{t+1}, ..., z_{t+T}|z_t, a)$, as detailed in Section 3. The generator operates autoregressively $g(z_{t+1}|z_t, a)$, and future states are obtained by iterative rollout. Due to the complexity of generating a full 3D latent state at once, we adopt a two-step pipeline: a Single Dynamics Model $SD$ that forecasts the coarse geometry of the next state, and a Latent Generator $LG$ that fills in detailed feature representations. Together, they construct the next latent $z_{t+1}$, which is then fed forward for rollout.

**Data Preparation.** Each training sequence is represented as $(z_1, \ldots, z_T, a)$. For a robot task, we uniformly sample $T$ intermediate timesteps as subgoals. At each $t$, multi-view images are processed by a pre-trained encoder (Xiang et al., 2025) to obtain a 3D latent $z_t$. During training, we randomly choose $t \in \{1, \ldots, T-1\}$ and use $(z_t, z_{t+1}, a)$ pairs for supervision.

### 4.1.1 SINGLE DYNAMICS MODEL

The single dynamics model $SD(\{\boldsymbol{p}_i\}_{t+1}|z_t, a)$ focuses on the dynamics, which predicts the sparse voxels of the next state conditioned on the current latent and the text instruction.

**Modeling.** We use conditional flow matching (Lipman et al., 2022) for generative modeling , which is closely related and largely equivalent in formulation to standard diffusion/score-matching objectives. Here, we adopt flow matching for simplicity and consistency in our setup. The voxel grid $\{0, 1\}^{N^3}$ is first encoded by 3D convolutional blocks and compressed into a latent tensor $\mathbb{R}^{N_c^3}$ with lower resolution ($N_c < N$). A transformer denoiser then operates in this compressed space.

**Conditioning.** Text instructions are encoded with a pre-trained CLIP model (Radford et al., 2021). The current latent $z_t$ is processed by 3D convolutions and aligned to resolution $N_c$. Both conditions share positional encodings with the voxel tokens, enabling the model to capture correlations, and are injected in each transformer block through cross-attention. To improve robustness to partial observations, we use condition augmentation, randomly dropping out voxel features from the input latent condition $z_t$ and adding Gaussian noise to its features $\{\boldsymbol{f}_i\}$.

### 4.1.2 LATENT GENERATOR

The latent generator $LG(\{\boldsymbol{f}_i\}_{t+1}|\{\boldsymbol{p}_i\}_{t+1}, z_t, a)$ predicts voxel features for the structure given by $SD$. Unlike $SD$, $LG$ focuses on appearance and visual details rather than dynamics. Similar to $SD$, it adopts a flow-matching framework with a transformer backbone, conditioned on text and 3D latent features via cross-attention. With this design, SD and LG can be trained separately but applied iteratively: SD predicts voxel positions, and LG fills in their features, producing complete 3D latents $z_{t+1}, ..., z_{t+T}$ over time.

## 4.2 PLANNING WITH INVERSE DYNAMICS

The 4D latent world model serves as the core of a robotic planner. Given a text instruction $a$ and the current state latent $z_0$, the model predicts future states $z_1, \ldots, z_T$ describing how the agent will interact with the environment.

**Goal-Conditioned Inverse Dynamics.** To translate predicted latents into robot control, we use a goal-conditioned inverse dynamics module: $ID(s_1, ..., s_H|z_t, z_{t+1})$, which outputs an action se-

quence $(s_1, \ldots, s_H)$, absolute joint positions representing low-level robot commands, from the current state $z_t$ to the subgoal $z_{t+1}$. Since this model does not perform long-horizon planning, it does not require the full details of the latent. Instead, we decode $z_t$ and $z_{t+1}$ into lighter point cloud representations $pc_t$ and $pc_{t+1}$. Then, a pyramid convolutional encoder (Ze et al., 2024a) to extract features from the point clouds, which are concatenated with robot joint states and passed to a diffusion head to predict the action sequence. To support different horizons, we randomly vary the action length during training and truncate or repeat them to a fixed horizon $H$. The inverse dynamics module is trained independently of the world model.

**Planning Pipeline.** The complete planning process is summarized in Algorithm 1. Starting from initial observations, the world model generates subgoals $z_1, \ldots, z_T$. The inverse dynamics model then predicts action sequences to reach each subgoal $z_{t+1}$, repeatedly replanning as needed. For closed-loop planning, latents can be updated from new observations after executing actions. This integration enables the 4D latent world model to serve as a planner for diverse robotic tasks.

---

**Algorithm 1** 4D Latent World Model for Robot Planning

---

1: Observe initial multi-view images $\{o_0^{(i)}\}$.
2: Encode initial state: $z_0 \leftarrow \mathcal{E}(\{o_0^{(i)}\})$.
3: Generate future 3D latents $\{z_1, ..., z_T\}$ conditioned on $z_0$ and instruction $a$.
4: **for** $t = 0, ..., T - 1$ **do**
5:    **while** agent has not reached subgoal $z_{t+1}$ **do**
6:       Decode latent to point cloud $pc_t \leftarrow \mathcal{D}(z_t)$ and $pc_{t+1} \leftarrow \mathcal{D}(z_{t+1})$.
7:       Predict action chunk with inverse dynamics model $s_1, ..., s_H \leftarrow ID(pc_t, pc_{t+1})$.
8:       Agent execute $H_a \leq H$ actions $s_1, ..., s_{H_a}$.
9:       **if** close loop planning **then**
10:          Observe new multi-view images $\{o_{t+1}^{(i)}\}$.
11:          Update next state: $z_{t+2} \leftarrow LG(SD(\mathcal{E}(o_{t+1}^{(i)})))$
12:       **end if**
13:    **end while**
14: **end for**

---

## 5 EXPERIMENTS

We evaluate the proposed 4D latent world model on both generation quality and downstream robot planning. Our experiments are designed to answer the following key questions:

- **4D Generation Quality:** How well does our model generate future 3D structures compared to state-of-the-art video-based and 4D world models, in terms of visual quality, physical consistency, and viewpoint invariance?
- **Robot Planning Performance:** Can the generated 3D structures be effectively used for robot planning, and how does our approach perform on complex manipulation tasks compared to baseline methods?

### 5.1 EXPERIMENTAL SETUP

**Training Data for 4D Latent World Model.** We collect training data from various robot planning tasks in ManiSkill3 (Tao et al., 2025) and LIBERO (Liu et al., 2023) simulators. Each task is paired with a language instruction and a set of successful trajectories. For ManiSkill3, we generate 1,000 demonstrations per task, and for LIBERO-90, 50 demonstrations per task. From each trajectory, we uniformly sample 4–10 intermediate timesteps and render multi-view observations using 40 cameras distributed spherically around the scene. To focus on relevant regions, we remove the background outside a pre-defined task-specific bounding box. Following the data preparation pipeline described in Section 4.1, each demonstration is converted into a standardized format for training the world model.

**Inverse Dynamics for Robot Planning.** We evaluate robot planning on three ManiSkill3 tasks: StackCube-v1, PullCubeTool-v1, and PegInsertionSide-v1 (Tao et al., 2025). For each task, we collect 1,000 demonstration trajectories with point cloud observations aggregated from four cameras,

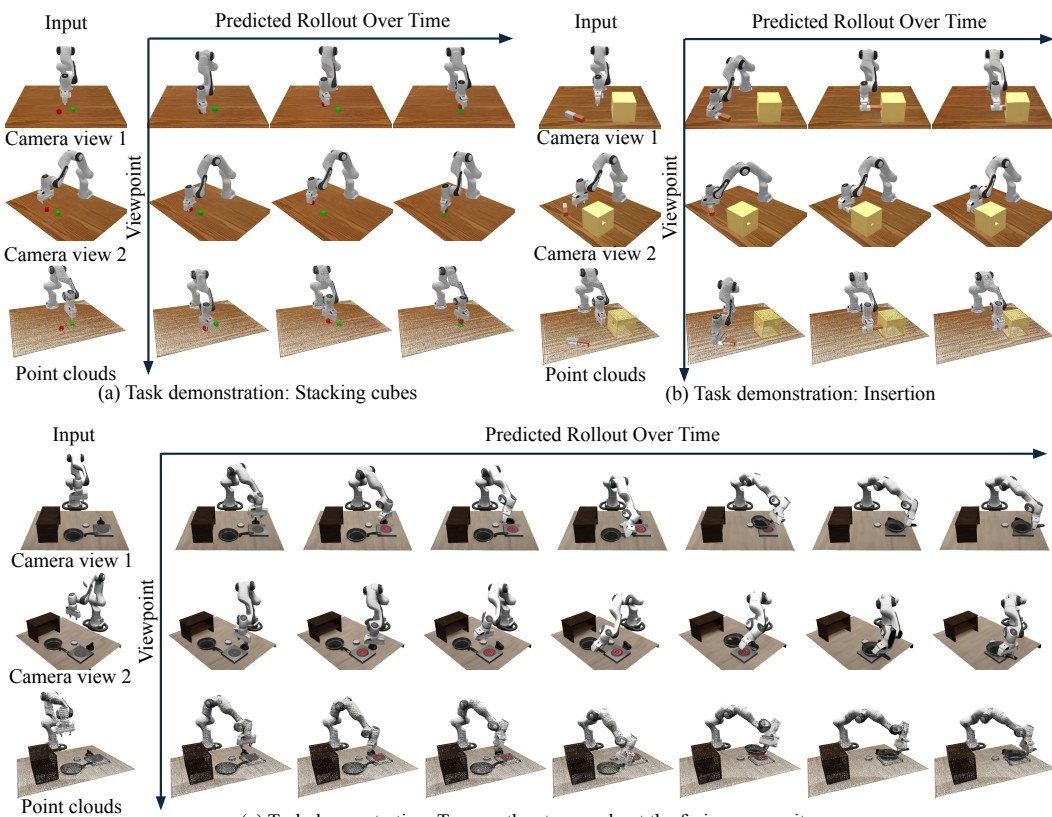

Figure 3: **4D generation visualizations.** Given input observations in the first column, our model unrolls the 4D latent world to generate future 3D structures over time. The first two rows show renderings from different camera viewpoints, and the third row shows corresponding point cloud visualizations. Text instructions for each task: (a) Pick up a red cube and stack it on top of a green cube, and let go of the cube without it falling. (b) Pick up an orange-white peg and insert the orange end into the box with a hole in it. (c) Turn on the stove and put the frying pan on it.

paired with corresponding action sequences. These demonstrations are used to train the inverse dynamics module (Section 4.2), which converts generated subgoals into executable actions. Evaluation is performed on the same tasks under different random initial conditions. The three tasks require varying levels of 3D understanding: StackCube-v1 involves stacking one cube on another, PullCubeTool-v1 requires using an L-shaped tool to pull a distant cube, and PegInsertionSide-v1 demands precise 3D alignment to insert a peg into a hole. The latter is especially sensitive to fine geometric accuracy; for this task, we relax the success clearance from 0.003 to 0.01.

**Baselines.** We compare our 4D generation and robot planning ability with the following baselines:

- **UniPi** (Du et al., 2023b) is a video planner that generates a single video about the robot manipulation trajectory and leverages inverse dynamics to get the robot control signal. Here we finetune a **Wan 2.1** (Wan et al., 2025) video generation model as the video planner.
- **TesserAct** (Zhen et al., 2025) is a 4D embodied world model that generates a sequence of paired RGB, depth, and normal, which enables robot manipulation.
- **OpenSora** (Zheng et al., 2024) is an image-to-video generation model, which is regarded as a baseline in world modeling generation comparisons.
- **Diffusion Policy** (Chi et al., 2023) and **3D Diffusion Policy** (Ze et al., 2024b) are state-of-the-art imitation learning methods.

UniPi and TesserAct are first finetuned on the same dataset to generate the video trajectory for each task. Then, an image-based diffusion inverse dynamics module is used to convert the generated video plan to robot actions. DP and DP3 are trained on expert demos for each task.

Table 1: **Evaluation of 4D generation.** We collect 5 key frames per trajectory and 40 camera views per frame, and evaluate both standard image quality metrics (PSNR, SSIM, LPIPS) and 3D consistency metrics from MVGBench (Xie et al., 2025b) (Chamfer Distance, depth error, cPSNR, cSSIM, cLPIPS). Compared to Wan-2.1, TesserAct, and OpenSora, our method achieves the best results on most metrics, with especially large improvements in 3D consistency, owing to its explicit 3D latent representation.

| | PSNR ↑ | SSIM ↑ | LPIPS ↓ | CD ↓ | depth ↓ | cPSNR ↑ | cSSIM ↑ | cLPIPS ↓ |
|---|---|---|---|---|---|---|---|---|
| Wan-2.1 | 19.87 | 0.84 | 0.09 | 43.09 | 25.06 | 16.86 | 0.62 | 0.24 |
| TesserAct | 21.63 | **0.86** | **0.07** | 42.79 | 23.87 | 17.91 | 0.65 | 0.23 |
| OpenSora-1.3 | 19.89 | 0.82 | 0.09 | 44.07 | 25.82 | 16.67 | 0.60 | 0.25 |
| Ours | **22.45** | 0.79 | 0.13 | **5.95** | **9.38** | **27.42** | **0.86** | **0.07** |

Figure 4: **Novel view generalization.** All models were trained on fixed global views but tested on a novel local viewpoint. Our model generates a consistent 3D scene from an unseen view, outperforming baselines significantly.

## 5.2 4D GENERATION RESULTS

**Visual Quality.** We begin by demonstrating the 4D generation capabilities of our proposed latent world model. Given multi-view images of the initial frame as input, our model autoregressively generates a sequence of future 3D latents to simulate the task's completion. Figure 3 visualizes the generated rollouts for several tasks. For each trajectory, we render the predicted 3D latents as images from two camera views and as a global point cloud. The results demonstrate that our generated 3D sequences maintain physical plausibility and temporal consistency while exhibiting high visual fidelity.

**Multiview Consistency.** Video generation-based approaches UniPi, TesserAct, and OpenSora, struggle to effectively integrate multi-view information. A common strategy for these models is to generate independent video sequences for each viewpoint and then attempt to fuse them at each timestep. However, without explicit 3D constraints, the independently generated views tend to lose consistency over time, which hinders the ability to leverage this multi-view information for downstream tasks, such as robot planning. In contrast, our model directly generates a unified 3D latent representation, which inherently enforces a consistent 3D structure and thus guarantees multi-view consistency by design. Table 1 presents a quantitative comparison against fine-tuned Wan 2.1, TesserAct, and OpenSora 1.3. As shown in the table, our method significantly outperforms the video-based approaches for multiview consistency.

**Viewpoint Generalization.** Many real-world tasks, particularly in mobile manipulation, cannot rely on fixed sensors and require observations from varying viewpoints. In such scenarios, it is crucial for a world model to generalize to novel viewpoints when simulating planning trajectories. Figure 4 demonstrates our model's robust ability to integrate diverse multi-view information and generalize to previously unseen viewpoints.

## 5.3 ROBOT PLANNING RESULTS

We evaluate our proposed 4D latent world model as a task planner, extracting actions at each step using a learned, goal-conditioned inverse dynamics model introduced in Section 4.2. We compare

Table 2: **Success rate for robot manipulation tasks.** Average success rate over 100 episodes, using four global cameras for observation. For PegInsertionSide-v1, the success clearance is relaxed to 0.01.

| | StackCube-v1 | PullCubeTool-v1 | PegInsertionSide-v1* | Average |
|---|---|---|---|---|
| DP | 56% | 87% | **24%** | 55.7% |
| DP3 | 47% | **94%** | 7% | 49.3% |
| UniPi | 9% | 5% | 1% | 5.0% |
| TesserAct | 13% | 1% | 3% | 5.7% |
| Ours | **84%** | 84% | 16% | **61.3%** |

Table 3: **Zero-shot generalization with visual and viewpoint changes.** Success rates on the StackCube-v1 task under unseen conditions at test-time. Perturbations include reduced lighting, additive Gaussian noise, a background color shift (R-channel change for table), and horizontal camera rotations ($5°$, $10°$).

| | Lighting | Noise | Background color | Viewpoint (5%) | Viewpoint (10%) |
|---|---|---|---|---|---|
| DP | 7% | 5% | 1% | 43% | 25% |
| DP3 | 47% | 47% | 47% | 49% | 45% |
| Ours | **78%** | **80%** | **84%** | **85%** | **83%** |

our method's manipulation performance against world modeling baselines UniPi and TesserAct, and state-of-the-art imitation learning policies DP and DP3. As shown in Table 2, our method significantly outperforms the video-based world models and achieves performance comparable to the specialized imitation learning policies. It is worth noting that the original DP3 implementation does not use color information for better generalization ability, which prevents it from distinguishing between colored objects in the StackCube-v1 task. More robot planning results can be found in Appendix B.1.

**Zero-shot generalization to visual and viewpoint changes.** Zero-shot generalization to novel visual conditions and camera views is critical for deploying robotic policies in real-world, unstructured environments. Some recent works (Zhu et al., 2024) have mentioned this point with some studies explicitly evaluating robustness to such changes. As demonstrated in Section 5.2, our model uses an explicit 3D latent representation, which naturally provides robustness to viewpoint changes. We now evaluate the policy's zero-shot planning performance under various perturbations, including changes in lighting, background color, additive image noise, and camera viewpoint. The results in Table 3 show that our method maintains a high success rate across these visual changes, demonstrating strong zero-shot generalization ability.

## 5.4 ABLATION STUDY

**Inputs to the Inverse Dynamics Module.** To understand the design choices of the inverse dynamics model, we evaluate three input types: using (i) our default downsampled point cloud (ii) the full 3D latents, and (iii) the 3D voxels as input to the inverse dynamic module. Due to the heavy computation complexity for 3D latents encoding with large number of camera views, here we use 4 cameras for inverse dynamics module training. Table 4 shows the success rate for robot task, which demonstrates the downsampled point cloud achieves performance comparable to latent-based inputs, providing a strong and lightweight signal for predicting robot actions.

**Number of camera views.** We train world models with 4, 10, and 40 camera views while keeping inference to 4 views. Planning success and 3D consistency improve with additional training views (Table 5 and Table 6), but even the 4-view model substantially outperforms video-based baselines. This shows the method remains effective under limited multi-view supervision.

## 5.5 REAL WORLD EXPERIMENTS

To evaluate the real-world applicability of our model, we collected a dataset of 100 human demonstrations for a physical block-in-basket task using five RGB cameras. From each trajectory, we uniformly sampled five intermediate frames and encoded them into 3D latents representation using the pre-trained encoder to form the training set. Figure 5 (a) illustrates our data collection setup.

Table 4: Ablation on inputs to the inverse dynamics module: success rate on the StackCube-v1 task.

| Point cloud (40 cams) | Point cloud (4 cams) | 3D Latents (4 cams) | 3D voxels (4 cams) |
|---|---|---|---|
| **84** % | 57 % | 59% | 40% |

Table 5: Ablation on the number of training-time camera views: visual consistency metrics on the StackCube-v1 task. All models use 4 cameras at inference time.

| | PSNR ↑ | SSIM ↑ | LPIPS ↓ | CD ↓ | depth ↓ | cPSNR ↑ | cSSIM ↑ | cLPIPS ↓ |
|---|---|---|---|---|---|---|---|---|
| Wan-2.1 | 20.10 | 0.84 | 0.09 | 38.74 | 24.54 | 16.95 | 0.62 | 0.22 |
| TesserAct | 22.26 | **0.87** | **0.06** | 39.11 | 24.86 | 17.75 | 0.64 | 0.22 |
| Ours (4 cams) | 19.81 | 0.75 | 0.18 | 7.10 | **8.81** | **28.89** | **0.86** | 0.07 |
| Ours (10 cams) | 21.78 | 0.77 | 0.14 | 7.06 | 9.85 | 26.92 | 0.85 | 0.07 |
| Ours (40 cams) | **22.39** | 0.78 | 0.13 | **6.81** | 9.98 | 26.75 | 0.85 | **0.07** |

We trained our 4D latent world model and inverse dynamics module on the collected real-world dataset, using the same configuration as in our simulation experiments. Figure 5 (b) and (c) illustrate qualitative generation results for real-world scenarios, while Figure 5 (d) presents visualizations of two policy rollouts. The generated point cloud sequences exhibit temporal and physical consistency, and the successful demonstrations indicate that our model learns meaningful dynamics from real-world data.

To quantitatively compare our approach with baselines, we randomly initialized object positions and evaluated our method against the Diffusion Policy (DP) over 50 episodes. Our method achieved a success rate comparable to DP (Ours 52%, DP 50%), demonstrating that our proposed model performs effectively in real-world robotic manipulation settings.

Table 6: Ablation on the number of training-time camera views: planning success rate on the StackCube-v1 task. The number of camera views refers to the world-model training setup; inference always uses 4 cameras.

| Ours (40 cams) | Ours (10 cams) | Ours (4 cams) | DP | DP3 | UniPi | TesserAct |
|---|---|---|---|---|---|---|
| **84%** | 72% | 57% | 56% | 47% | 9% | 13% |

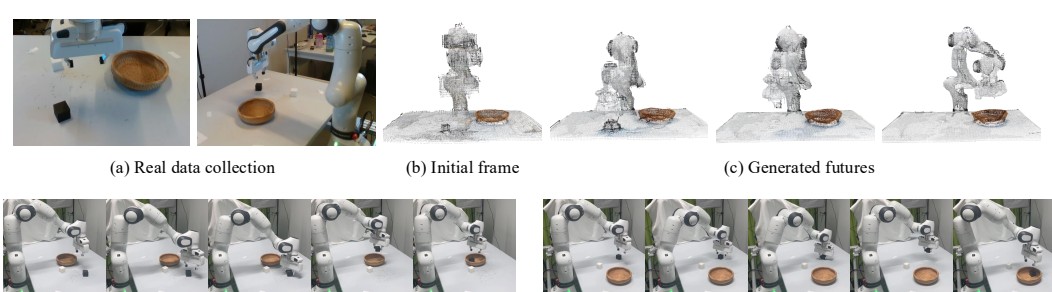

(a) Real data collection    (b) Initial frame    (c) Generated futures

(d) Real world execution

Figure 5: **Real world experiments.** We collect real robot data (a), reconstruct the initial input frame from these observations (b), predict future rollouts in real environment (c), and execute proposed policy at test time (d).

## 6 CONCLUSION

We have introduced a *4D latent world model* for robot planning, which predicts the evolution of 3D scene structure directly in a compact latent space. By moving beyond prevailing 2D video-based approaches, our model learns a dynamic model in 3D latent space that encodes holistic scene structure to enforce 3D consistency, producing rollouts of future latents that can be decoded into explicit formats such as point clouds or rendered views. Integrated with a goal-conditioned inverse dynamics module, these latents serve as geometrically grounded subgoals that translate into executable actions. Our experiments demonstrate that this approach achieves state-of-the-art performance in 3D-aware generative modeling, yielding significant improvements in downstream robotic planning tasks. While our current implementation assumes calibrated multi-view inputs to reconstruct the initial latent, extending to weaker input settings is a promising direction for broader applicability.

ETHICS STATEMENT

The authors have considered the ethical implications of this research and have found no direct ethical concerns. Our work is foundational, focusing on improving the planning capabilities of robotic agents. The data used for training and evaluation is sourced from established public robotics benchmarks, and our small-scale real-world data collection did not involve sensitive information or raise privacy concerns. We believe our work adheres to the ICLR Code of Ethics.

REPRODUCIBILITY STATEMENT

We are committed to ensuring the reproducibility of our work. We plan to release our full implementation after a thorough code cleanup and documentation process. All simulated data used in our experiments was generated using the official, publicly available APIs of the ManiSkill and LIBERO benchmark suites. To ensure fair and robust comparisons, all reported metrics and success rates were obtained using fixed random seeds and consistent evaluation environments.

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

## THE USE OF LARGE LANGUAGE MODELS

Large Language Models (LLMs) were used to assist with manuscript language polishing and literature search. All substantive research content and scientific conclusions are the original work of the authors.

## A   IMPLEMENTATION DETAILS

### A.1   4D WORLD MODEL TRAINING

Our framework consists of two main components: a single dynamics model and a latent generator, which were trained independently. Both were implemented as conditioned flow matching models following the architecture proposed by Xiang et al. (2025). Here, we extend the original conditions to both text and 3D latent. The latent generator operates on a $64 \times 64 \times 64$ voxelized grid with a feature dimension of $d = 8$, while the dynamics model uses a similar architecture on a coarser $16 \times 16 \times 16$ grid. Both models consist of 24 transformer blocks, each with 16 attention heads and a model dimension of 1024. For text conditioning, we utilized embeddings from a pre-trained CLIP model. The dynamics model is also conditioned on the input 3D latent; we use sparse 3D convolutional layers to match the latent's resolution to the model's internal dimension, after which it is injected into the cross-attention blocks together with the text condition.

Training for both models spanned 300,000 steps with a learning rate of $1 \times 10^{-4}$. We used a per-GPU batch size of 8 with mixed-precision (FP16) computation. For the latent generator, we applied 4 gradient accumulation steps, and for the dynamics model, we used 2 steps. The optimizer used was AdamW with no weight decay. To enable classifier-free guidance, we set the unconditional dropout probability to 0.1 and applied an exponential moving average (EMA) with a decay rate of 0.9999 to stabilize the training. Each model was trained for approximately 3 days on four NVIDIA H100 (80GB) GPUs.

### A.2   INVERSE DYNAMICS MODEL TRAINING

Our goal-conditioned inverse dynamics model is trained on 1,000 expert demonstrations for each task. The policy is formulated as a diffusion model that takes point clouds representing the current and goal states as input. For observation encoding, we employ a point cloud encoder introduced by Ze et al. (2024a), which comprises four 1D convolutional layers with a hidden dimension of 128. The action decoder is a 1D conditional UNet with downsampling channel dimensions of [256, 512, 1024], a kernel size of 5, and 8 groups for normalization layers.

The inverse dynamics model was trained for 20,000 epochs. At inference time, we use 100 denoising steps to predict the action sequence. Training for each task took approximately 8 hours on a single NVIDIA A100 GPU.

### A.3   VIDEO WORLD MODEL BASELINE FINE-TUNING

To establish our baselines, we utilize Wan 2.1 (Wan et al., 2025) as the video generative backbone of UniPi (Du et al., 2023b), and the TesserAct (Zhen et al., 2025) generative model. Both models were fine-tuned on the same dataset. For the TesserAct model specifically, we generated depth maps alongside the images from the simulator and used its official codebase to create normal maps. For fine-tuning, Wan 2.1 was trained for 10,000 steps on two NVIDIA H100 GPUs, achieving convergence in approximately 36 hours. TesserAct was also fine-tuned for 10,000 steps on four NVIDIA H100 GPUs, taking around two days to converge.

## B   ADDITIONAL EXPERIMENT RESULTS

### B.1   ROBOT PLANNING RESULTS

In Section 5.3, we compare our proposed methods with baselines TesserAct and UniPi. However, the success rate of both methods in ManiSkill3 tasks are very low. To provide a more rigorous evalua-

tion of robot planning capabilities, we extended our experiments to the RLBench benchmark, where TesserAct (Zhen et al., 2025) and UniPi (Du et al., 2023b) have established performance records. We evaluated our method on three RLBench tasks: *CloseBox*, *SweepToDustpan*, and *WaterPlants*, collecting 1,000 demonstrations for each. We utilized 20 cameras for model training and 4 cameras for inference, maintaining the same 4D world model and inverse dynamics architecture described in Section 5.3. For the baselines, we cite the success rates reported in the official TesserAct publication (Zhen et al., 2025). For our method, we report the success rate averaged over 100 random episodes per task. Table 7 shows the success rate comparison in 3 RLBench tasks.

Table 7: **Success rate on RLBench.** Average success rate over 100 episodes for our model. For Image-BC, UniPi, and TesserAct baselines, the success rate is from Zhen et al. (2025).

|  | Close Box | Sweep To Dustpan | Water Plants | Average |
|---|---|---|---|---|
| Image-BC | 53% | 0% | 0% | 17.7% |
| UniPi | 81% | 49% | 35% | 55.0% |
| TesserAct | 88% | 56% | 41% | 61.7% |
| Ours | **93%** | **69%** | **64%** | **75.3%** |

## B.2 4D GENERATION RESULTS

In Section 5.2, we compare the performance of 4D generation results with video generation based baselines. We have shown that our proposed model performs a strong ability in multiview consistency and viewpoint generalization, while maintaining the comparable visual quality at the same time. In order to further evaluate the connection of world modeling quality and robot policy performance more directly, we use Segment Anything Model 3 (SAM3) (Carion et al., 2025) to segment the robot shape in each generated image, and compare the IoU score of robot mask between generation and ground truth. Table 8 shows the IoU score comparison with OpenSora-1.3 (Zheng et al., 2024), Wan-2.1 (Wan et al., 2025), and TesserAct (Zhen et al., 2025), which shows that our proposed model can provide a stable and accurate generation for robot planning.

Table 8: **IoU score of robot mask.** We collect 5 key frames per trajectory and 40 camera views per frame, and evaluate the IoU between ground truth robot mask and generated robot mask.

|  | StackCube-v1 ↑ | PullCubeTool-v1 ↑ | PegInsertionSide-v1 ↑ | Average ↑ |
|---|---|---|---|---|
| Wan-2.1 | 0.7423 | 0.7189 | 0.7148 | 0.7253 |
| TesserAct | 0.8302 | 0.7704 | 0.7741 | 0.7915 |
| OpenSora-1.3 | 0.7789 | 0.6956 | 0.6945 | 0.7229 |
| Ours | **0.9091** | **0.9334** | **0.8970** | **0.9132** |

