# OpenReview forum: "4D Latent World Model for Robot Planning"
_ICLR.cc/2026/Conference — Submitted to ICLR 2026_

### Official Review · Reviewer_XFnZ · 2025-10-27

**Soundness:** 3
**Presentation:** 3
**Contribution:** 2
**Rating:** 4
**Confidence:** 3

**Summary:**

This paper proposes a 4D latent world model to predict the evolution of 3D scenes in a latent representation. The 4D latent world model integrates a Single Dynamics Model (SD) and a Latent Generator Model (LG), designed to forecast changes in 3D voxels and their corresponding features. Leveraging this 4D latent world model, the authors develop a robot planning framework that utilizes predicted point clouds for future forecasting. Qualitative and quantitative results demonstrate the effectiveness of the proposed method on ManiSkill3 and LIBERO.

**Strengths:**

- The concept of a 4D world model is promising and has garnered significant interest within the research community.
- The authors propose a technically sound framework by decomposing the challenges of the 4D world model into coarse geometry prediction and feature prediction.
- The authors develop a robot planning framework and demonstrate that the proposed 4D world model enhances the performance of robot planning.
-The visual results validate the effectiveness of the proposed method in simulated environments.

**Weaknesses:**

Major weakness:
- Although the authors acknowledge that the current method is limited to surface-level projections (L5, L82), the proposed 4D world model still relies on surface points (L192), which are back-projected into a 3D representation.
- The concept of the 4D latent world model appears unclear. The fundamental representation of the world model is a voxel grid, with each voxel corresponding to a latent feature. Consequently, the proposed 4D latent world model is not entirely "latent."
- While the authors state that "3D reasoning becomes apparent in high-precision manipulation" (L52), the Single Dynamics Model (SDM) only supports coarse predictions (as shown in Figure 5). It is unclear how this supports high-precision manipulation.

Minor weakness:
- Although the Latent Generator Model (LG) predicts latent features, the robot planning framework solely utilizes point clouds from the SDM. This diminishes the significance of predicting future latent features.

**Questions:**

- The rendering results appear impressive in synthetic environments. How do the real-world rendering results perform, and does the method suffer from a significant sim-to-real gap?
- Is it possible to incorporate future latent predictions into the robot planning framework, instead of relying solely on point clouds?
- How are the feature embeddings from different views aggregated (L192)?
- The training details for the sparse encoder (L192) and sparse decoder (L194) are not provided. If my understanding is correct, this procedure aims to map DINO features to 3D Gaussians. Why not directly map RGB to 3D Gaussians?

---

> ### Author Response · Authors · 2025-11-22
>
> We thank the reviewer for the valuable feedback. Below are our responses to each comment and a description of the clarifications and extra results now present in the **revised manuscript** and our [**website**](https://iclr2026-4553.github.io/).
>
> 1. **Proposed method is still surface-level**: Thank you for the comment. By surface-level, we refer to approaches that reconstruct 3D from RGB images paired with depth or normal maps, which remain fundamentally 2.5D and operate in pixel space. In contrast, our method constructs a full 3D representation of the entire scene: multi-view features are back-projected into a voxel grid and aggregated to form an intrinsic 3D latent that can be decoded into point clouds or rendered from arbitrary viewpoints. This enables inherent 3D modeling rather than relying solely on 2.5D projections. We have added clarification in **Sec. 2** of the manuscript.
> 2. **Voxel grid with latent features is not entirely latent**: Thank you for this observation. Our voxel grid is a lightweight latent representation of the scene. It contains roughly 8k active voxels, each with an 8-dimensional feature vector (i.e., 8000 $\times$ 3 voxel coordinates and 8000 $\times$ 8 latent features). Although the voxel coordinates provide explicit 3D position, the feature vectors themselves constitute the latent representation, similar to how positional encodings are used in 2D VAE-based latents. Compared to the latents used in video world models (e.g., VAE feature maps of  \#images$\times$height$\times$width$\times$C), our latent is far more compact while still carrying the necessary 3D structure. It is also significantly smaller than explicit 3D representations such as NeRFs or 3D Gaussian fields. Therefore, the voxel grid with latent features is compact, structured, and much lighter than explicit geometry, while retaining explicit positional cues needed for consistent 3D modeling.
> 3. **SDM only supports coarse predictions, how does this support high-precision manipulation**: Thank you for raising this point. SDM is designed to capture the coarse structural motion of the scene, analogous to a high-level planner that determines where and how objects should move. LG then conditions on both the SDM output and the previous 3D latent to recover the fine-grained geometric details. The full 3D latent produced by SDM+LG encodes detailed 3D information, which can be decoded to dense point clouds and also serve as conditioning for future dynamics. In this way, SDM provides global dynamics, while LG restores the fine-scale structure needed for precise manipulation and future dynamics.
> 4. **Planning use only SDM point clouds, diminishing the role of LG**: Thank you for identifying this point. The planning framework does not rely on the coarse SDM prediction alone. We decode the full 3D latent produced by SDM+LG into a dense point cloud and then downsample it for inverse-dynamics input. LG is important as it refines the SDM output with fine-grained geometry and appearance cues, which ensures accurate future latents and reliable control. Planning uses the SDM+LG decoded point cloud, not SDM alone. 3D voxels generated by SDM is a coarse representation of the scene, which is not enough for accurate robot manipulation. We also add an **ablation study** for comparing different types of input for inverse dynamics module, which is shown at the response 6.
> 5. **Real-world rendering results perform and sim-to-real gap**: Thank you for noting this point. The 3D Gaussian rendering result for real-world case does not match the visual quality seen in simulation, due to the limited camera views. However, the decoded point clouds shown at **Figure 5** are meaningful, which can be used for inverse dynamics module in robot task. We have updated video demos for real-world experiments on the [**website**](https://iclr2026-4553.github.io/#real-world-experiments), which demonstrates our proposed world modeling method performs well in the real world settings.

---

> ### Author Response · Authors · 2025-11-22
>
> 6. **Can future latent predictions be incorporated directly into planning, instead of using point clouds**: Thank you for bringing up this concern. For the input of inverse dyamics module, we use point clouds rather than latent features because (i) point clouds provide sufficient geometric structure for control while being significantly lighter-weight, (ii) this design decouples the inverse dynamics from the world model, allowing it to be trained modularly and to generalize across tasks, and (iii) encode dense 3D latents for dense timestep is way more costly than acquire the point cloud (it is acceptable for world model training because we only need 3D latents for several sub-goals, but inverse dynamics training needs dense 3D latents at each timestep). We also include an **ablation study** in the revised paper that conditions the inverse dynamics directly on latent features. We can see that the inverse dynamics based on the light-weight point clouds achieves similar results as full 3D latents. Here is the ablation results for different inputs (success rate for StackCube-v1 task)
>     | Point cloud (40cam) | Point cloud (4cam) | 3D Latents (4cam) | 3D voxels (4cam) |
>     | - | - | - | - |
>     | **84\%** | 57\% | 59\% | 40\% |
>
>     Note that it is computationally heavy to prepare the training data for a large number of camera views for 3D latents, here we use 4 cameras as our ablation setting.
> 7. **How the feature embeddings from different views aggregated**: Thank you for catching this. As described in the paper, we construct the 3D latent by first extracting patch-level DINOv2 features from each input view, unprojecting them into the 3D voxel grid using the known camera intrinsics and extrinsics, and then aggregating features that fall into the same voxel (e.g., via averaging). The aggregated per-voxel features are subsequently refined by a sparse 3D encoder to produce the final latent representation. This unprojection and fusion strategy is a standard approach for multi-view 3D feature integration, and our implementation follows the formulation used in TRELLIS for constructing SLat. We have added a clarification in **Sec. 3.2** in the paper.
> 8. **training details for the sparse encoder/decoder and why not map RGB directly to 3D Gaussians**: Thank you for identifying this concern. We follow the latent construction module used in TRELLIS. The sparse encoder and sparse decoder correspond to the pre-trained 3D latent VAE from TRELLIS, which was trained using RGB reconstruction losses (L1, D-SSIM, LPIPS) between rendered Gaussians and the ground-truth images. We have added a clarification in **Sec. 3.2** of the paper. DINO features provide higher-level image embeddings that can be unprojected into a common 3D voxel grid using camera, allowing us to aggregate them into a coherent latent representation before decoding.

---

> > ### Comment · Reviewer_XFnZ · 2025-11-27
> > **Response to authors**
> >
> > The rebuttal has addressed some of my concerns, and I appreciate the time and effort put into it.
> >
> > However, my main concern remains that the key technique of this paper is a hybrid representation implemented using an explicit grid filled with latent features. This constitutes a significant difference, as the paper claims to present a "4D Latent" world model.
> >
> > The authors could better position their work by providing a clearer explanation of the key components (such as point cloud input, grid structure, and latent representation) which would make this paper a stronger submission.

---

> > > ### Author Response · Authors · 2025-11-28
> > > **Response to Reviewer XFnZ**
> > >
> > > We thank the reviewer for the continued engagement during the rebuttal and the insight into the definition of our "latent" representation. We have already updated the manuscript and the **new change is in blue**.
> > >
> > > We agree that the difference between a global unstructured latent (like a 1D vector in VAE) and "**structured grid latent**" we adopted is important and needs clarification. To address this, we have revised the Abstract, Introduction, and Formulation sections in the manuscript and explicitly define our representation as a "structured 3D latent", specifically, a sparse voxel grid of compressed latent features.
> > >
> > > We believe this representation qualifies as a "latent" world model for the following key reasons supported by recent literature, which we have already clarified in the paper:
> > > 1. **Analogy to 2D Latent Diffusion**: In state-of-the-art image or video generation approaches (e.g., Latent Diffusion Models [1]), the "latent" is typically a downsampled 2D feature map $ (C, H, W) $. They are compressed representations of the original pixel space which are considered as latents. For recent SOTA video generation models such as Wan 2.1 [2], they operate on the 2D latents with temporal dimension $(C, T, H, W)$, where they downsample spatially by $8\times$ and temporally by $4\times$. Similarly, our representation functions as a *compressed latent for 3D space*. It encodes the 3D scene into a low dimensional feature space $C=8$ in a sparse grid $ 64 \times 64 \times 64 $ [3], rather than operating on raw high dimentional explicit geometry like dense meshes or radiance fields.
> > > 2. **Compress Rate**: Our representation is highly compressed. As noted in the revised **Sec. 3.2**, we utilize approximately 8K active sparse voxels with 8 dimensional features, which is approximately 80K floats (including coordinates and features). Our representation achieves a significant compression ratio compared to the dense $64^3$ grid or the explicit 3DGS parameters which often requires millions of parameters [4].
> > > 3. **Clear Explanation of the Key Components**: As described in **Sec. 3.2**, structured 3D latent we adopted is composed of a set of sparse voxels $z = \{(p_i, f_i)\}_{i=1}^L$. Here, $p_i \in [N]^3$ represents the discrete coordinates within a sparse $64 \times 64 \times 64$ voxel grid, and $f_i \in \mathbb{R}^8$ is the compressed latent feature vector for that voxel. Crucially, the spatial structure $\{p_i\}$ is highly sparse, containing only around 8K active voxels. This compact latent $z$ acts as a generative seed that can be decoded into dense, explicit 3D representations, such as 3DGS or dense point clouds ($\sim$ 200K points).
> > >
> > > We fully agree that it is important to clarify the definition of "latent" in our paper. We have revised the paper to use "structured 3D latent" term and highlight this hybrid design with explicit sparse grid voxels and latent feature embeddings. We believe this structure is important to enable the model to maintain the 3D consistency and physical consistency which unstructured 1D latents lack.
> > >
> > > [1] Robin Rombach, Andreas Blattmann, Dominik Lorenz, Patrick Esser, and Bjorn Ommer. High-resolution image synthesis with latent diffusion models. In *Proceedings of the IEEE/CVF conference on computer vision and pattern recognition*, pp. 10684–10695, 2022.
> > >
> > > [2] Team Wan, Open and advanced large-scale video generative models. *arXiv preprint arXiv:2503.20314*, 2025.
> > >
> > > [3] Jianfeng Xiang, Zelong Lv, Sicheng Xu, Yu Deng, Ruicheng Wang, Bowen Zhang, Dong Chen, Xin Tong, and Jiaolong Yang. Structured 3d latents for scalable and versatile 3d generation. In *Proceedings of the Computer Vision and Pattern Recognition Conference*, pp. 21469–21480, 2025.
> > >
> > > [4] Bernhard Kerbl, Georgios Kopanas, Thomas Leimkuhler, and George Drettakis. 3d gaussian splatting for real-time radiance field rendering. *ACM Transactions on Graphics*, 42(4), July 2023.

---

### Official Review · Reviewer_fvTr · 2025-10-28

**Soundness:** 2
**Presentation:** 3
**Contribution:** 2
**Rating:** 4
**Confidence:** 3

**Summary:**

This paper introduces 4D Latent World Model, a framework that learns to model and predict 4D trajectories—that is, the evolution of 3D scene representations over time—in a latent space. The model comprises two components: a Single Dynamics (SD) module that captures coarse voxel-level occupancy dynamics, and a Latent Generator (LG) module that focuses on finer-grained visual appearance and local geometry. The outputs from SD and LG are fused and converted into point-cloud representations, which are then processed by a diffusion-based inverse-dynamics head to generate low-level action sequences from the high-level latent plan.

For evaluation, the authors compare their work to Wan-2.1 and Tesseract on generative quality, and to both of them and Diffusion Policy (DP) and DP3 on a single robotic manipulation task.

**Strengths:**

1. With a 4D world-model operating as planning backbone, they show improvements in robustness against visual and viewpoint changes.
2. The paper is well written and easy to follow.

**Weaknesses:**

While the paper makes an interesting conceptual argument that injecting 3D semantics into video world models can improve visuomotor planning and robotic control, the experimental evidence does not sufficiently support this claim. In its current form, the evaluation appears limited in choice of baselines and tasks.

1. The paper primarily compares to UniPi and TesserAct, which are relatively weak references for assessing world-model quality. To substantiate the argument that 3D semantics improve video world models, comparisons to stronger baselines such as Video Latent Diffusion Model (Video LDM), OpenSora, or DreamerV3 would be more appropriate.

2. The proposed planning approach closely resembles Decision Diffuser, but the experiments are restricted to only three ManiSkill3 tasks, with near-zero performance from competing baselines. This makes it unclear whether the reported results reflect genuine improvements or task-specific tuning. A fairer and more comprehensive evaluation would include additional manipulation benchmarks where the baselines have reported their results, like RLBench (UniPi and TesserAct), Kuka stacking (Decision Diffuser), Meta-World.

3. It would be informative to include an ablation comparing the flow-matching objective with the standard diffusion objective to clarify how the chosen training formulation affects model performance.

4. The real-world experiments are presented only qualitatively. Reporting success rates or other quantitative results would provide stronger evidence that the learned 4D world model improves real-robot control.

5. It would be helpful to include an ablation demonstrating whether utilizing the full latent representation z contributes more to the final performance compared to using a simplified representation p.

Overall, the experimental design does not yet convincingly demonstrate that injecting 3D semantics into video world models leads to measurable performance gains. With broader and more rigorous comparisons, this could become a stronger contribution.
I am open to adjusting my score if the authors can address my concerns during the rebuttal.

**Questions:**

Please see my questions in the above section.

---

> ### Author Response · Authors · 2025-11-22
>
> Many thanks for the constructive comments. We respond to each item below and have revised the **manuscript** and our [**website**](https://iclr2026-4553.github.io/) to include clarifications and extra results.
>
> 1. **Comparison with stronger video-generation baselines**: Thank you for noting this point. For the video generation backbone of UniPi and TesserAct, we are using state-of-the-art open-source video generation models, and we fine-tuned them on our robotics dataset. For the UniPi baseline, due to the limitations of its original video-generation backbone, we replaced it with the Wan2.1 T2V backbone (1.3B, released Feb 2025). For the TesserAct baseline, the underlying model is CogVideoX (5B, released Aug 2024). Table 1 shows the visual comparison between Wan-2.1, TesserAct, and ours, where the fine-tuned Wan and TesserAct have strong video generation quality. However, because these models are directly modeling 2D pixel space, while ours is directly modeling 3D structure, there is a gap in 3D multi-view generation consistency. Regarding the suggested alternatives: VideoLDM has not released training code for conditional video prediction; and DreamerV3 is not a video-generation backbone, but rather a different modeling approach that is not directly comparable to our setup. We attempted to fine-tune the OpenSora 1.3 model but found it does not work well in our settings. To clarify, we currently finetune the T2V model plus an image condition head because it allows for rapid experimentation. We show the comparison of generation results on our [**website**](https://iclr2026-4553.github.io/#baselines-comparison). Also, we have started to finetune the I2V model and will report the visual metrics soon.
> 2. **Evaluation on more benchmarks**: Thank you for raising this point. We agree that broader evaluation across additional manipulation benchmarks would strengthen the work. Currently we are setting up the experiments in other Benchmarks such as RLBench, where TesserAct and UniPi report the success rate. We plan to add the comparison in the final version of manuscript.
> 3. **Flow-matching and diffusion comparison**: We thank the reviewer for pointing this out. Conceptually, flow-matching and the common diffusion/score-matching formulations are closely related, and the training objective are basically the same. They both specify a continuous transport between data and a simple noise distribution and learn vector fields that reverse that transport. In practice the distinction often reduces to whether one directly matches a score (denoising objective) or a velocity field (flow-matching), and many recent works show formal connections via the probability flow ODE / reverse-time SDE viewpoint. Because of this close connection, we opted to present results using the flow-matching objective and focus the paper on architecture and application. We have added a discussion in **sec. 4.1.1** clarifying the relationship between the two approaches and our implementation.
> 4. **Real-world results**: Thanks for pointing this out. We have already updated video demos for real-world robot task on the [**website**](https://iclr2026-4553.github.io/#real-world-experiments) and updated **Sec. 5.5** in the paper, which verifies our proposed world model pipeline is suitable and stable in real-world settings. However, the real world setup is complex, making it difficult to report comprehensive quantitative numbers at this stage.
> 5. **Ablation on different representation in planning**: We thanks the reviewer for pointing this out. For the input of inverse dyamics module, we use point clouds rather than latent features because (i) point clouds provide sufficient geometric structure for control while being significantly lighter-weight, (ii) this design decouples the inverse dynamics from the world model, allowing it to be trained modularly and to generalize across tasks, and (iii) encode dense 3D latents for dense timestep is way more costly than acquire the point cloud (it is acceptable for world model training because we only need 3D latents for several sub-goals, but inverse dynamics training needs dense 3D latents at each timestep). We also include an **ablation study** in the revised paper that conditions the inverse dynamics directly on latent features. We can see that the inverse dynamics based on the light-weight point clouds achieves similar results as full 3D latents. Here is the ablation results for different inputs (success rate for StackCube-v1 task)
>     | Point cloud (40cam) | Point cloud (4cam) | 3D Latents (4cam) | 3D voxels (4cam) |
>     | - | - | - | - |
>     | **84\%** | 57\% | 59\% | 40\% |
>
>     Note that it is computationally heavy to prepare the training data for a large number of camera views for 3D latents, here we use 4 cameras as our ablation setting.

---

> ### Author Response · Authors · 2025-12-04
> **Updates of the Response to Reviewer fvTr**
>
> We thank the reviewer for their valuable suggestions and we have updated the experimental results in the **revised manuscript**.
>
> 1. **More video-generation baselines**: We have already added the comparison of the OpenSora baseline. Here, we use OpenSora 1.3 (1B, 2025) I2V base model and finetune it with our robotic dataset. We have already updated the comparison of generation visual metrics in **Table 1**:
>     |   | PSNR $ \uparrow $ | SSIM $ \uparrow $ | LPIPS $ \downarrow $ | CD $ \downarrow $ | depth $ \downarrow $ | cPSNR $ \uparrow $ | cSSIM $ \uparrow $ | cLPIPS $ \downarrow $ |
>     | - | - | - | - | - | - | - | - | - |
>     | Wan-2.1 | 19.87 | 0.84 | 0.09 | 43.09 | 25.06 | 16.86 | 0.62 | 0.24 |
>     | TesserAct | 21.63 | **0.86** | **0.07** | 42.79 | 23.87 | 17.91 | 0.65 | 0.23 |
>     | OpenSora-1.3 | 19.89 | 0.82 | 0.09 | 44.07 | 25.82 | 16.67 | 0.60 | 0.25 |
>     | Ours | **22.45** | 0.79 | 0.13 | **5.95** | **9.38** | **27.42** | **0.86** | **0.07** |
>
>     and the comparison of IoU score of the robot mask in **Appendix B.2**:
>     | | StackCube-v1 $ \uparrow $ | PullCubeTool-v1 $ \uparrow $ | PegInsertionSide-v1 $ \uparrow $ | Average $ \uparrow $ |
>     | - | - | - | - | - |
>     | Wan-2.1 | 0.7423 | 0.7189 | 0.7148 | 0.7253 |
>     | TesserAct | 0.8302 | 0.7704 | 0.7741 | 0.7915 |
>     | OpenSora-1.3 | 0.7789 | 0.6956 | 0.6945 | 0.7229 |
>     | Ours | **0.9091** | **0.9334** | **0.8970** | **0.9132** |
>
> 2. **Evaluation on more benchmarks**: We constructed training datasets for 3 tasks in RLBench using the same data collection method as in our paper. We evaluated our method and compared the average success rate with TesserAct and UniPi, based on the table reported in TesserAct paper. We have also added this analysis to **Appendix B.1**.
>     | | Close Box | Sweep To Dustpan | Water Plants | Average |
>     | - | - | - | - | -|
>     | UniPi | 81\% | 49\% | 35\% | 55.0\%|
>     | TesserAct | 88\% | 56\% | 41\% | 61.7\% |
>     | Ours | **93\%** | **69\%** | **64\%** | **75.3\%** |
>
> 3. **Real-world results**: We have already updated video demos for real-world robot task on the [**website**](https://iclr2026-4553.github.io/#real-world-experiments) and updated **Sec. 5.5** in the paper, which verifies our proposed world model pipeline is suitable and stable in real-world settings. For quantitative evaluation, we conducted 50 episodes with randomly initialized object positions. Our method achieved a success rate of 52%, compared to 50% for the Diffusion Policy. This performance is comparable to state-of-the-art imitation learning approaches, demonstrating that our method is effective in real-world robotic settings.

---

### Official Review · Reviewer_2THB · 2025-11-01

**Soundness:** 2
**Presentation:** 2
**Contribution:** 2
**Rating:** 4
**Confidence:** 4

**Summary:**

This paper proposes learning a world model over 3D representations to enable robot planning with generalization capability. Instead of relying on explicit 3D representations, the authors use a latent space that can be rendered into point clouds or 3D Gaussian splatting. In the experiments, the proposed approach is compared with several baselines and shows superior performance in multiple simulated tasks.

**Strengths:**

This paper addresses an important research question and provides comparisons against several baselines. The learned dynamics model is also evaluated through real-world rollouts.

**Weaknesses:**

1: Learning the dynamics for robot planning over 3D representations has been extensively studied. The authors primarily discuss world models based on video prediction but should also address prior work that learn dynamics over 3D representations (e.g., [1-7]).  Please clarify the main conceptual and methodological differences between this paper and those works.

2: What’s the definition of robot action in this paper? The paper uses multiple terms such as “instruction”, “task plan”, and “joint states”. Please clarify what’s the actual interface to the robot?

3: Related to my last point, there are missing details about the inverse dynamics module. Why is the full latent representation not required? One motivation for learning latent representations is to encode diverse features efficiently. How is the inverse dynamics model trained-what data, supervision, and objective functions are used?

4: In section 4.2, the paper mentions “closed-loop planning”, what’s the control frequency, and under what conditions is replanning triggered?

5: In tables 1 and 2, the baselines sometimes outperform the proposed approach. Please discuss the insights behind these results.

6: Could the inverse dynamics model be used to enable closed-loop control in real-world experiments? This ties back to the definition of robot actions. Additionally, what frequency can the system achieve for closed-loop planning in the real world?

7: What are the failures modes of the proposed approach? How diverse is the training dataset, and to what extent can the model generalize to out-of-distribution scenarios.

8: For the planning module, it would be valuable to include an ablation study on other planning approaches (e.g., sampling-based or gradient-based methods) to better understand its role and why learning an inverse dynamics model is necessary.

References:

[1]: H. Chen, Y. Niu, K. Hong, S. Liu, Y. Wang, Y. Li, and K. R. Driggs-Campbell, “Predicting object interactions with behavior primitives: An application in stowing tasks,” in 7th Annual Conference on Robot Learning, 2023.

[2]: Y. Huang, C. Agia, J. Wu, T. Hermans, and J. Bohg, “Points2plans: From point clouds to long-horizon plans with composable relational dynamics,” in 2025 IEEE International Conference on Robotics and Automation (ICRA), 2025.

[3]: H. Jiang, H.-Y. Hsu, K. Zhang, H.-N. Yu, S. Wang, and Y. Li. Phystwin: Physicsinformed reconstruction and simulation of deformable objects from videos. arXiv preprint arXiv:2503.17973, 2025.

[4]: H. Shi, H. Xu, S. Clarke, Y. Li, and J. Wu, “Robocook: Longhorizon elasto-plastic object manipulation with diverse tools,” in 7th Annual Conference on Robot Learning, 2023.

[5]: H. Shi, H. Xu, Z. Huang, Y. Li, and J. Wu, “Robocraft: Learning to see, simulate, and shape elasto-plastic objects in 3d with graph networks,” The International Journal of Robotics Research.

[6]: Y. Huang, N. C. Taylor, A. Conkey, W. Liu, and T. Hermans, “Latent Space Planning for Multi-Object Manipulation with EnvironmentAware Relational Classifiers,” IEEE Transactions on Robotics (T-RO), 2024.

[7]: D. Driess, Z. Huang, Y. Li, R. Tedrake, and M. Toussaint, “Learning multi-object dynamics with compositional neural radiance fields,” in Conference on robot learning.

**Questions:**

See the Weaknesses section.

---

> ### Author Response · Authors · 2025-11-22
>
> We appreciate the reviewer’s careful reading and useful suggestions. Below are our point-by-point responses and the corresponding revisions added to the **manuscript** and our [**website**](https://iclr2026-4553.github.io/).
>
> 1. **Relation to prior 3D dynamics works**: We thank the reviewer for pointing out these relevant works. We have added a discussion of [1–7] in the **revised Related Work** section (highlighted in blue). Prior 3D-dynamics approaches typically operate on explicit geometric structures, e.g., segmented point clouds, relational graphs, particle systems, or multi-object NeRFs. These are often designed for task-specific regimes (e.g., deformable-object manipulation, relational multi-object reasoning). Their planning and dynamics modules are usually separate and require predefined object primitives or action spaces. In contrast, our method learns a single holistic latent 3D scene state that aggregates multi-view inputs, captures full-scene geometry, and supports 4D dynamics entirely in latent space. This latent is compact, expressive, and decodable into multiple 3D formats (e.g., point clouds, 3D Gaussians), enabling planning directly through latent rollouts conditioned only on language instructions. This formulation yields a flexible, task-agnostic world model without requiring object segmentation, or hand-engineered action primitives. These differences are now clearly described in the revision.
> 2. **Clarification of robot actions**: We thank the reviewer for the question. Our planning system involves **two action interfaces**. First, the latent world model is conditioned on a *high-level language instruction* (e.g., “stack the cubes”, “insert the peg”), which specifies the intended evolution of the scene and guides the 4D latent rollout. Second, the inverse dynamics module converts the predicted futures into *low-level robot control signals* by outputting a sequence of absolute joint positions for each timestep. Thus, the actual interface to the robot is a trajectory of joint positions, while language instructions serve only as high-level conditioning for the world model. We have added this clarification in **Sec. 3.1** and **Sec. 4.2**.
> 3. **Details of inverse dynamics module**: Thank you for identifying this concern. The inverse dynamics module takes the current and predicted future downsampled point clouds, decoded from our 3D latents, and uses a PointNet encoder followed by a diffusion head to predict an action segment of length 32 or 64. The action is defined as absolute joint positions, and the training objective is a diffusion loss. We use point clouds rather than latent features because (i) point clouds provide sufficient geometric structure for control while being significantly lighter-weight, (ii) this design decouples the inverse dynamics from the world model, allowing it to be trained modularly and to generalize across tasks, and (iii) encoding dense 3D latents for every timestep is significantly more costly than acquiring point clouds (it is feasible for world model training because we only need 3D latents for several sub-goals, but inverse dynamics training needs dense 3D latents at each timestep). We also include an **ablation study** in the revised paper that conditions the inverse dynamics directly on latent features. We can see that the inverse dynamics based on the light-weight point clouds achieves similar results as full 3D latents. Here is the ablation results for different inputs (success rate for StackCube-v1 task)
>     | Point cloud (40cam) | Point cloud (4cam) | 3D Latents (4cam) | 3D voxels (4cam) |
>     | - | - | - | - |
>     | **84\%** | 57\% | 59\% | 40\% |
>
>     Note that it is computationally heavy to prepare the training data for a large number of camera views for 3D latents, here we use 4 cameras as our ablation setting.
> 4. **Closed-loop planning**: Thank you for pointing this out. "Closed loop" here means our method can replan with the 3D latent world model after each action chunk (e.g., 32 or 64 action steps). The model can either generate 3D latents for all subgoals at once (as open loop), or regenerate every next subgoal after the previous one is reached (as closed loop). We currently evaluate closed-loop planning exclusively in simulation.

---

> ### Author Response · Authors · 2025-11-22
>
> 5. **Baselines sometimes outperform ours**: Thank you for noting this. In Table 1, video-based baselines occasionally outperform our method on 2D image metrics such as SSIM or LPIPS. This is expected, since these models operate entirely in pixel space and explicitly optimize for 2D appearance consistency. In contrast, our model learns a full 3D latent representation of the scene, which leads to significantly better 3D consistency (CD, depth, cPSNR, cSSIM, cLPIPS) and multi-view coherence, both of which are much more important for downstream robot planning. In Table 2, our method clearly outperforms world-model baselines (UniPi and TesserAct) on all manipulation tasks. Compared to strong imitation-learning policies, our approach achieves comparable or better success rates, especially on geometry-sensitive tasks that benefit from accurate 3D reasoning. These results show that although 2D baselines may score slightly higher on isolated pixel-space metrics, our model produces 3D-consistent rollouts that translate into substantially stronger planning performance.
> 6. **Closed-loop control in real-world experiments**: Thank you for pointing this out. Currently we only test closed-loop planning in the simulators. In the real world demos (updated on the [**website**](https://iclr2026-4553.github.io/#real-world-experiments)), we use open-loop planning where the subgoals are generated at once. After subgoals are generated, the inverse dynamics module converts them to absolute joint positions which is used to control the robot.
> 7. **Failure modes**: Failures primarily occur in 2 ways: (1) inaccurate 3D generation, which can occur in fine-grained, high-precision tasks such as peg insertion, and (2) inverse dynamics prediction errors. **Training dataset diversity**: The training set combines demonstrations from ManiSkill and LIBERO. **Generalization to out-of-distribution scenarios**: As shown in the paper (Tables 2–3), the model generalizes well to significant camera viewpoint changes and visual perturbations such as lighting shifts, background color changes, and additive image noise. These results demonstrate strong robustness to out-of-distribution visual conditions.
> 8. **Other planning approaches**: Thank you for the insightful point. We appreciate the suggestion. Other planning approaches like sampling or gradient based methods would serve as valuable baselines, and we plan to run some of them (like CEM or MPPI) and add into the final version of the manuscript.

---

### Official Review · Reviewer_7oiN · 2025-11-02

**Soundness:** 3
**Presentation:** 3
**Contribution:** 2
**Rating:** 4
**Confidence:** 4

**Summary:**

The paper proposes a 4D latent world model for robot planning that predicts how a 3D scene evolves over time from multi‑view images and a text instruction. The core idea is to encode a scene into a sparse 3D voxel latent and roll it forward with two modules: a Single Dynamics (SD) model for coarse geometry and a Latent Generator (LG) for appearance/features.

On simulated manipulation tasks in ManiSkill3 and LIBERO, the method reports strong multi‑view 3D consistency and competitive task success relative to video‑based planners and imitation baselines. Qualitative real‑world rollouts from 4 RGB‑D cameras are presented.

**Strengths:**

Overall it's well motived. It shows multi‑view consistency improvements. Also large gains on 3D metrics supporting the main claim that modeling directly in a 3D latent enforces consistency across views.

Planning performance and robustness. In simulation we see it outperforms the two video‑based world model baselines and is comparable to strong imitation policies; robustness under lighting/background/camera shifts is also solid.

**Weaknesses:**

Might need some analysis that connects world modeling quality to policy performance. Likely the accuracy of small region is more important than general image distance metrics.

Missing qualitative real world experiments.

**Questions:**

Is there particular motivation of using the sparse latent representation inspired by SLAT?
The contraction needs 40 camera views which seems unrealistic in real. Any ablation on the number of camera views?
What if you do inverse dynamics on the latents (before using LG to fill in detailed feature representations). This ablation seems important to understand the value of the proposed design

---

> ### Author Response · Authors · 2025-11-22
> **Response to Reviewer 7oiN**
>
> We thank the reviewer for the constructive feedback. Below we address each concern and have incorporated clarifications and additional results into the **revised manuscript** and our [**website**](https://iclr2026-4553.github.io/).
>
> 1. **Analysis that connects world modeling quality to policy performance**: Thanks for pointing this out. Currently, we compare the visual quality, 3D consistency and robot task success rate with several baselines, which verifies the 3D accuracy and consistency for robot policies. Moreover, in order to quantify the generation results for robot manipulation more directly, we use Segment Anything Model 3 (SAM3) to segment the robot shape in each generated image, and compare the **IoU score** of robot mask between generation and ground truth. We summarize the IoU metrics as follows:
>     | | StackCube-v1 $\uparrow$ | PullCubeTool-v1 $\uparrow$ | PegInsertionSide-v1 $\uparrow$ | Average $\uparrow$ |
>     | - | - | - | - | - |
>     | Wan-2.1 | 0.7423 | 0.7189 | 0.7148 | 0.7253 |
>     | TesserAct | 0.8302 | 0.7704 | 0.7741 | 0.7915 |
>     | OpenSora-1.3 | 0.7789 | 0.6956 | 0.6945 | 0.7229 |
>     | Ours | **0.9091** | **0.9334** | **0.8970** | **0.9132** |
> 2. **Ablation on the number of camera views**: Thank you for raising this concern. We use 40 cameras only during world-model training to obtain high-quality multi-view supervision and improve 3D latent reconstruction. During inference on robot-manipulation tasks, we use only 4 cameras, matching realistic sensor setups. In the real environment, it is feasible to acquire richer multi-view information by spinning the camera around the workspace and record video, which can be used to reconstruct the 3D scene. Moreover, we added an ablation varying the number of training-time cameras (4, 10, 40) in the paper at Sec 5.4. In visual consistency metrics, however, the 10-camera model performs reasonably close to the 40-camera model, and even the 4-camera model remains substantially stronger than video-based baselines. For the success rate for robot task, fewer cameras training also outperforms baselines. These results indicate that while additional views improve latent 3D fidelity, the world model remains effective even under significantly reduced multi-view supervision. The ablation results for visual metrics across different camera views are shown below (metrics on StackCube-v1 task):
>     |   | PSNR $\uparrow$ | SSIM $\uparrow$ | LPIPS $\downarrow$ | CD $\downarrow$ | depth $\downarrow$ | cPSNR $\uparrow$ | cSSIM $\uparrow$ | cLPIPS $\downarrow$ |
>     | - | - | - | - | - | - | - | - | - |
>     | Wan-2.1 | 20.10 | 0.84 | 0.09 | 38.74 | 24.54 | 16.95 | 0.62 | 0.22 |
>     | TesserAct | 22.26 | **0.87** | **0.06** | 39.11 | 24.86 | 17.75 | 0.64 | 0.22 |
>     | Ours (4cam) | 19.81 | 0.75 | 0.18 | 7.10 | **8.81** | **28.89** | **0.86** | 0.07 |
>     | Ours (10cam) | 21.78 | 0.77 | 0.14 | 7.06 | 9.85 | 26.92 | 0.85 | 0.07 |
>     | Ours (40cam) | **22.39** | 0.78 | 0.13 | **6.81** | 9.98 | 26.75 | 0.85 | **0.07** |
>
>     Here is the success rate for StackCube-v1 task, where 4 camera views are used during inference:
>     | Ours (40cam) | Ours (10cam) | Ours (4cam) | DP | DP3 | UniPi | TesserAct |
>     | - | - | - | - | - | - | - |
>     | **84\%** | 72\% | 57\% | 56\% | 47\% | 9\% | 13\% |
> 3. **Inverse dynamics on the latents**: Thank you for raising this question. For the input of inverse dyamics module, we use point clouds rather than latent features because (i) point clouds provide sufficient geometric structure for control while being significantly lighter-weight, and (ii) this design decouples the inverse dynamics from the world model, allowing it to be trained modularly and to generalize across tasks. We clarify that prior to LG statge, we only have sparse voxels, which may not be accuate enough for robot manipulation. We also include an **ablation study** in the revised paper that conditions the inverse dynamics directly on latent features. We can see that the inverse dynamics based on the light-weight point clouds achieves similar results as full 3D latents and 3D voxels. Note that it is computationally intensive to prepare the training data for a large number of camera views for 3D latents, here we use 4 cameras for our ablation setting. Here is the ablation results for different inputs (success rate for StackCube-v1 task)
>     | Point cloud (40cam) | Point cloud (4cam) | 3D Latents (4cam) | 3D voxels (4cam) |
>     | - | - | - | - |
>     | **84\%** | 57\% | 59\% | 40\% |
> 4. **Real world experiments**: Thanks for the suggestion. We have already updated the video demos for a real-world robot task on our [**website**](https://iclr2026-4553.github.io/#real-world-experiments), which demostrates that our proposed world model pipeline is suitable and stable in real-world settings.

---

### Author Response · Authors · 2025-12-03
**Final Response (Part 2/2)**

### 3. Comparisons with baselines and connection to planning performance (Reviewer 7oiN, 2THB, fvTr)
Reviewers requested comparisons with stronger models and clearer evidence that our improved 3D structure leads to better planning.
We emphasize that UniPi and TesserAct already use strong, state-of-the-art video generation backbones in our implementation, Wan 2.1 T2V (1.3B, 2025) and CogVideoX (5B, 2024), which we fine-tuned on our robotics dataset. As shown in both the main paper and the comparison figures on our project webpage, these models produce high-quality video generations. We also added the results from fine-tuned OpenSora 1.3 I2V (1B, 2025) suggested by Reviwer fvTr. The results are as follows and have also been added to **Table 1** in the paper.

|   | PSNR $ \uparrow $ | SSIM $ \uparrow $ | LPIPS $ \downarrow $ | CD $ \downarrow $ | depth $ \downarrow $ | cPSNR $ \uparrow $ | cSSIM $ \uparrow $ | cLPIPS $ \downarrow $ |
| - | - | - | - | - | - | - | - | - |
| Wan-2.1 | 19.87 | 0.84 | 0.09 | 43.09 | 25.06 | 16.86 | 0.62 | 0.24 |
| TesserAct | 21.63 | **0.86** | **0.07** | 42.79 | 23.87 | 17.91 | 0.65 | 0.23 |
| OpenSora-1.3 | 19.89 | 0.82 | 0.09 | 44.07 | 25.82 | 16.67 | 0.60 | 0.25 |
| Ours | **22.45** | 0.79 | 0.13 | **5.95** | **9.38** | **27.42** | **0.86** | **0.07** |

The experiments show that our proposed 4D latent world model achieves substantially better 3D consistency metrics (CD, depth, cPSNR, cSSIM, cLPIPS) which are the metrics most relevant for 3D spatial reasoning and robot planning, and maintain the comparable 2D pixel metrics (e.g., SSIM/LPIPS) with video generation baselines at the same time.

To further connect world-model quality to downstream control (Reviewer 7oiN’s request), we added a SAM3-based robot-mask IoU metric that directly measures the accuracy of predicted robot motion. Our model achieves a significantly higher IoU than Wan 2.1, TesserAct, and OpenSora 1.3, strengthening the argument that improved 3D structure leads to more reliable predicted futures and higher planning success.
| | StackCube-v1 $ \uparrow $ | PullCubeTool-v1 $ \uparrow $ | PegInsertionSide-v1 $ \uparrow $ | Average $ \uparrow $ |
| - | - | - | - | - |
| Wan-2.1 | 0.7423 | 0.7189 | 0.7148 | 0.7253 |
| TesserAct | 0.8302 | 0.7704 | 0.7741 | 0.7915 |
| OpenSora-1.3 | 0.7789 | 0.6956 | 0.6945 | 0.7229 |
| Ours | **0.9091** | **0.9334** | **0.8970** | **0.9132** |

### 4. More robot planning benchmarks (Reviewer fvTr)
Reviewers requested more results on robot planning benchmarks, especially for RLBench where TesserAct reported success rates. We constructed training datasets for 3 tasks in RLBench using the same data collection method as in our paper. We evaluated our method and compared the average success rate with TesserAct and UniPi, based on the table reported in TesserAct paper. We have also added this analysis to **Appendix B.1**.
| | Close Box | Sweep To Dustpan | Water Plants | Average |
| - | - | - | - | -|
| UniPi | 81\% | 49\% | 35\% | 55.0\%|
| TesserAct | 88\% | 56\% | 41\% | 61.7\% |
| Ours | **93\%** | **69\%** | **64\%** | **75.3\%** |


Finally, we added (a) an **ablation** on the number of training camera views in **Sec. 5.4** (Reviewer 7oiN), (b) additional discussion to prior 3D dynamics works in **Sec. 2** (Reviewer 2THB), and (c) clarifications on the structured sparse voxel latent space and its relation to prior 3D latent work in **Sec. 3.2** (Reviewer XFnZ). Reviewer-specific responses are provided in our replies to each reviewer. We believe these additions address all reviewer concerns comprehensively.

---

### Author Response · Authors · 2025-12-03
**Final Response (Part 1/2)**

We appreciate AC taking on the responsibility of reviewing our submission under these unusual circumstances. We thank all reviewers for their thoughtful feedback on our submission. Reviewers described the paper as “well-motivated,” “technically sound,” and “addresses an important research question,” and noted that our 4D latent world model shows “improvements in robustness against visual and viewpoint changes” and “large gains on 3D metrics.”
We have carefully addressed all concerns, added new quantitative evaluations, clarified key design choices, and revised the **manuscript** with highlighted changes. We also expanded our project [**website**](https://iclr2026-4553.github.io/) with real-world demonstrations and comparisons. Below we summarize how we addressed the main shared concerns across reviewers.

### 1. Real-world experiments (Reviewer 7oiN, 2THB, fvTr, XFnZ)
Several reviewers asked about our real-world evaluation and the sim-to-real gap.
We have already updated video demos for real-world robot task on the [**website**](https://iclr2026-4553.github.io/#real-world-experiments) and updated **Sec. 5.5** in the paper, which verifies our proposed world model pipeline is suitable and stable in real-world settings. For quantitative evaluation, we conducted 50 episodes with randomly initialized object positions. Our method achieved a success rate of **52\%**, compared to **50\%** for the Diffusion Policy. This performance is comparable to state-of-the-art imitation learning approaches, demonstrating that our method is effective in real-world robotic settings.


### 2. Using point clouds for inverse dynamics module (Reviewer 7oiN, 2THB, fvTr, XFnZ)
Reviewers raised questions about the design choice to feed point clouds, rather than 3D latents, into the inverse dynamics module.
For the input of inverse dyamics module, we use point clouds rather than latent features because (i) point clouds provide sufficient geometric structure for control while being significantly lighter-weight, (ii) this design decouples the inverse dynamics from the world model, allowing it to be trained modularly and to generalize across tasks, and (iii) encode dense 3D latents for dense timestep is way more costly than acquire the point cloud (it is acceptable for world model training because we only need 3D latents for several sub-goals, but inverse dynamics training needs dense 3D latents at each timestep).
We also include an **ablation study** in the revised paper that conditions the inverse dynamics directly on latent features. We can see that the inverse dynamics based on the light-weight point clouds achieves similar results as full 3D latents. Here is the ablation results for different inputs (success rate for StackCube-v1 task).
| Point cloud (40cam) | Point cloud (4cam) | 3D Latents (4cam) | 3D voxels (4cam) |
| - | - | - | - |
| **84\%** | 57\% | 59\% | 40\% |

Note that it is computationally heavy to prepare the training data for a large number of camera views for 3D latents, here we use 4 cameras as our ablation setting.

---

### Meta-Review · Area_Chair_NSyS · 2026-01-02

**Summary:**

This work is on 4D latent world models capable of predicting how a 3D scenes evolves over time in the future, and their usage for planning of high-precision manipulation tasks. A trained dynamics module performs prediction followed by a generator module.

All four reviewers provided lukewarm ("4") ratings with no reviewer championing the paper throughout the reviewing process. Multiple issues were raised:
- Positioning wrt the SoTA (Reviewer 2THB provided a long list of references to similar work). The authors attempted an answer, but the AC fails to understand why the SoTA is more task-specific than the proposed method, as the rebuttal claims.
- Missing analysis connecting world modeling quality to policy performance: a table was provided by the authors is the rebuttal but surprinsgly was not commented by them.
- Lukewarm performance, with baselines outperforming the proposed method on several points (this seems to have been answered, these metrics are pixel based on not relevant).
- Overclaim of the model being latent while still being defined on a voxel grid. Planning only uses point clouds. This is one of the main criticism, to which the authors could not provide a (convincing) answer.
- No quantitative eval for the real world experiments.
- Overclaim on high-precision manipulation.
- Lacking clarity and missing details.
- Weak baselines and missing video world models.
- Problems with the experimental setup.
- Lacking analyses and discussions. No discussions of failure modes;

The authors attempted to answer all the issues raised by the reviewers, but in the AC's reading (and reviewer XFnZ's reading) were convincing only very partially on some minor points, which were the first ones in the list above. Substantial problems persisted, in particular positioning wrt. the SoTA, positioning of the approach (a "latent" model), choice of baselines and several overclaims.

The AC therefore judges that the paper does not fulfil the requirements for acceptance.Explained in the meta-review.

**Reviewer Concerns:**

Explained in the meta-review.

**Reviewer Scores:**

XFnZ noted that some of his concerns have been addressed but not the main one. They would have been unlikely to raise their score.

---

### Decision · Program_Chairs · 2026-01-26

Reject